# Carcinosarcoma of the Endometrium—Pathology, Molecular Landscape and Novel Therapeutic Approaches

**DOI:** 10.3390/medicina61071156

**Published:** 2025-06-26

**Authors:** Stoyan Kostov, Yavor Kornovski, Vesela Ivanova, Deyan Dzhenkov, Dimitar Metodiev, Mohamed Wafa, Yonka Ivanova, Stanislav Slavchev, Eva Tsoneva, Angel Yordanov

**Affiliations:** 1Research Institute, Medical University Pleven, 5800 Pleven, Bulgaria; drstoqn.kostov@gmail.com; 2Department of Gynecology, St. Anna University Hospital, Medical University of Varna, 9002 Varna, Bulgaria; ykornovski@abv.bg (Y.K.); yonka.ivanova@abv.bg (Y.I.); st_slavchev@abv.bg (S.S.); 3Department of General and Clinical Pathology, Faculty of Medicine, Medical University Sofia, 2 Zdrave Str., 1432 Sofia, Bulgaria; veselaivanovamd@gmail.com; 4Department of General and Clinical Pathology, Forensic Medicine and Deontology, Division of General and Clinical Pathology, Faculty of Medicine, Medical University—Varna “Prof. Dr. Paraskev Stoyanov”, 9002 Varna, Bulgaria; ddzhenkov@gmail.com; 5Clinical Pathology Laboratory, MHAT “Nadezda” Women’s Health Hospital, 1373 Sofia, Bulgaria; dimitarmetodievvv@gmail.com; 6Neuropathological Laboratory, University Hospital “Saint Ivan Rilski”, 1431 Sofia, Bulgaria; 7St. Louise Frauenklinik, 33098 Paderborn, Germany; m.wafa@vincenz.de; 8Department of Reproductive Medicine, Specialized Hospital for Active Treatment of Obstetrics and Gynaecology Dr Shterev, 1330 Sofia, Bulgaria; dretsoneva@gmail.com; 9Department of Gynecologic Oncology, Medical University Pleven, 5800 Pleven, Bulgaria

**Keywords:** endometrial carcinosarcoma, sarcomatous component, heterologous, molecular landscape, target therapies

## Abstract

Endometrial carcinosarcoma (ECS) is a rare and aggressive histological subtype of endometrial cancer that is associated with a dismal prognosis. It is a biphasic metaplastic carcinoma with a monoclonal origin comprising epithelial and mesenchymal components. The ECS originates from the epithelial components of the tumor, which undergoes an epithelial-to-mesenchymal transition. Approximately half of patients are diagnosed at the early stage of the disease, whereas the other half are diagnosed at the advanced stage. More than one-third of women present with metastatic lymph nodes, and approximately 10% will have distant metastases. Therefore, ECS is the deadliest type of endometrial cancer compared to other high-grade endometrial carcinomas. Surgical resection with adjuvant therapy remains the standard of care in most cases. The rarity of this disease hinders conducting prospective clinical trials to establish the optimal treatment regimens and increase overall survival. There are no specific guidelines for managing these rare and aggressive tumors despite the increasing interest in ECS in the gynecologic oncology community. The present review focuses on all new insights into ECS regarding its epidemiology, pathology, prognosis, and treatment. Furthermore, the molecular characteristics and new treatment regimens for primary (early and advanced stages) and recurrent ECS are discussed in detail.

## 1. Introduction

Endometrial carcinosarcoma (ECS), also known as malignant mixed Müllerian tumor, malignant mesodermal mixed tumor, or metaplastic carcinoma, is a rare, aggressive, high-grade endometrial cancer [1,2]. It is a biphasic metaplastic carcinoma composed of epithelial and mesenchymal elements, where the sarcoma element has dedifferentiated from the carcinoma element [3,4,5]. Dixon and Dockerty first reported on “carcinosarcomatodes of the uterus” in 1940 [6]. Since then, ECS has attracted considerable attention regarding its diagnosis, surgical management, and adjuvant treatment novelties. It accounts for approximately 5% of all uterine malignancies [2,5]. ECSs are aggressive malignant tumors with a dismal prognosis. It is diagnosed in an advanced stage more often compared to endometrial endometrioid cancer [5]. It is characterized by a high incidence of lymphatic spread, peritoneal dissemination, and hematogenous metastases [1]. ECSs are more aggressive tumors compared to high-grade endometrial carcinomas [7]. The rarity of this disease hinders conducting prospective clinical trials to establish the optimal treatment regimens and increase overall survival. Therefore, there are no specific guidelines for managing these rare and aggressive tumors despite increasing interest in ECS in the gynecologic oncology community. Recently, ESGO/ESTRO/ESP published guidelines for managing patients with endometrial carcinoma in which the treatment for ECS was briefly mentioned [8]. Recently, new insights into the molecular markers and different molecular types of ECSs have been elucidated [9,10]. The present review focuses on all new insights into ECS regarding its epidemiology, pathology, prognosis, and treatment. Furthermore, the molecular characteristics and new treatment regimens for primary (early and advanced stages) and recurrent ECS are discussed in detail.

## 2. Epidemiology

The annual incidence of ECS in patients has increased over the past two decades—the worldwide rate of growth in its incidence has increased by approximately 2% annually [5,11]. Brooks et al. conducted an analysis of 2677 women with uterine sarcomas during the period 1989–1999 (carcinosarcomas were classified as uterine sarcomas during that time). The authors found an incidence of ECS of 4.3 per 1,000,000 Afro-Americans vs. 1.7 per 1,000,000 Caucasian women and 0.99 per 1,000,000 women of other races [12]. In Europe, Boll et al. observed an increased rate of growth in ECS from 5.1 to 6.9 per 1,000,000 [13]. The proportion of ECS among all types of endometrial carcinomas has also increased from 1.7% to 5.6% [4,5,13,14,15]. Although they are rare tumors, ECSs represent 16.4% of uterine-malignancy-associated deaths [1]. The incidence of ECS is almost equal to that of some types of non-endometrioid endometrial cancers, such as clear cell cancer (2–4%) and undifferentiated (5%). However, ECS is significantly rarer than serous endometrial cancer (10%) [15,16,17]. Additionally, ECS is more common than mixed epithelial endometrial carcinoma (with an incidence of less than 3%). Mixed epithelial endometrial carcinoma is a malignant uterine disease in which two or more spatially distinct histological types are observed (at least one of which is serous or clear cell carcinoma) [18]. The incidence of ECS is higher compared to that of the most commonly observed sarcomas of the uterine corpus (leiomyosarcomas and low-grade endometrial stromal sarcomas—less than 2% of all malignancies of the uterine corpus) [19]. Carcinosarcomas of the female reproductive system represent 80% of all carcinosarcomas in the human population, followed by lung, bladder, and breast. Uterine corpus/cervix carcinosarcomas account for approximately 80% of carcinosarcomas in the genital system, whereas 18.2% occur in the fallopian tubes/ovaries [20]. Thoracic carcinosarcomas (mainly lung and breast) comprise 9.6% of all carcinosarcomas [20].

## 3. Risk Factors

Historically, ECS is considered a disease of the elderly, with peak incidence at approximately 70–79 years [4,21]. However, in recent years, a decrease in the age of the patients affected by this disease has been observed, where the most significant interval increase in the rates of its incidence has been demonstrated for women aged between 60 and 69 (an annual increase of 2.7%) [4,5,21]. Therefore, in recent years, the average age at ECS diagnosis has decreased to 67 years [4,21]. Generally, the incidence of ECS increases after approximately 50 years of age [21,22,23]. Women diagnosed with ECS are approximately 20 years older compared to the average age in women diagnosed with other uterine malignancies [14,21,22].

The overall incidence of ECS in Afro-American women is 2 to 3 times higher compared to that in Caucasian women. However, the largest interval increase in its incidence rates between 2000 and 2016 has been observed in Hispanic women [4,21,24]. Afro-American women diagnosed at an early stage of the disease experience earlier recurrence and lower progression-free and overall survival compared to those in Caucasian women with early-stage ECS. However, this disparity in the overall survival among races is not observed for women in the advanced stages of the disease [25]. Patients with ECS more often presented at an advanced stage of the disease compared to women with grade 3 endometrioid endometrial carcinoma [15].

Previous exposure to radiation therapy is also a risk factor, and women with ECS that has developed after previous radiation are younger in age compared to women with tumors arising de novo [23,26]. Pothuri et al. reported 23 women who developed post-radiation endometrial cancer, of whom 8 patients (35%) had ECS [27].

Tamoxifen use is also associated with the occurrence of ECS [23,28]. However, the incidence of ECS or uterine sarcoma development after tamoxifen use is lower compared to the incidence of type I endometrial cancer after the initiation of tamoxifen [23,29,30]. ECS shares some similar risk factors with endometrial cancer—a high body mass index, diabetes, hypertension, nulliparity, early-age menstruation, onset in post-menopausal age, and exogenous estrogen exposure [1,4,31]. However, these risk factors are not specific, and their effects on ECS occurrence do not correlate significantly compared to those seen for type I endometrial cancer [23].

It is suggested that Lynch syndrome, also known as hereditary nonpolyposis colorectal cancer, could be related to ECS [32,33]. Lynch syndrome is typically associated with type I endometrial cancer. However, non-endometrioid types, such as serous carcinoma, clear cell carcinoma, and carcinosarcoma of the uterine corpus, are also observed [33]. Existing studies have reported on the identification of a germline MLH1 mutation in genetic testing in women with ECS [32,33]. Cowden syndrome could also contribute to the development of ECS. The syndrome caused by mutations in the pentaerythritol tetranitrate (PTEN) gene is associated with an increased lifetime risk of breast and endometrial cancer [34,35,36,37].

Interestingly, an increased incidence of ECS among women with breast cancer has been observed. Wilson and Cordell analyzed 387 ECS patients from the Northern and Yorkshire Cancer Registry between 1998 and 2007. The researchers also observed 85,930 patients who may have developed ECS following breast cancer during the study period. In 22.5% (87 women) of ECS cases, the tumor constituted a second primary malignancy following breast cancer, and a time interval of 10–20 years for the occurrence of uterine malignancies was seen. The authors deliberately mentioned that these findings were found independently of hormone therapy, as it had no significant effect on the occurrence of ECS following breast cancer. This study identified two patients with germline BRCA1 mutations and the development of ECS. Although speculation, this finding suggests that ECS could be associated with hereditary ovarian breast cancer syndrome and particularly with BRCA1/BRCA2 carriers. However, the reason for the association between these two malignancies could be a yet-unknown genetic mutation [38].

## 4. Pathogenesis and Pathology

ECS consists of two populations of cell components—epithelial (carcinomatous) and mesenchymal (sarcomatous) components. ECS was traditionally regarded as a malignant mixed Müllerian tumor and has historically been described as one of the most aggressive uterine sarcomas since it is biphasic, consisting of epithelial and mesenchymal components. The classification of ECS has changed numerous times over the years. In 1906, Kheres proposed the term “mixed mesodermal tumor” [23,39]. In 1935, Mc Farland observed 119 cases of ECS and tried to establish the gross and microscopic characteristics of this tumor. Ober, in 1965, attempted to classify mesenchymal uterine sarcomas and define any tumor of the uterine corpus which consists of stromal sarcoma and one or more heterologous elements as a “mixed mesodermal tumor” [23,40]. The author also mentioned that a tumor could be classified as a “mixed mesodermal tumor” if it contained two or more heterologous elements [23,40]. Several pathogenic hypotheses for ECS were proposed in the past. The collusion theory states that the carcinoma and sarcoma components arise as separate tumors and later collide to form one malignancy. The combination theory suggests that both tumors originate from a single monoclonal stem cell, which undergoes separate differentiation in the initial steps of tumorigenesis. The composition theory hypothesized a potential pseudosarcomatous stromal reaction of the carcinomatous element [1,7,14,41,42,43]. However, the currently accepted theory is the conversion theory, which states that ECS originates from the epithelial components of the tumor, which undergoes an epithelial-to-mesenchymal transition. The neoplasm is regarded as a metaplastic carcinoma with a monoclonal origin. Therefore, currently, ECS is well recognized as a high-grade endometrial carcinoma [1,3,8,14].

The epithelial-to-mesenchymal transition was first described by Elizabeth Hay in 1968. The author defined it as a biological mechanism where epithelial cells can downregulate their epithelial characteristics and obtain mesenchymal characteristics [44,45]. Currently, the epithelial-to-mesenchymal transition is clarified as a process where the epithelial cells lose their polarity and cell–cell contact, reorganize the cell cytoskeleton, obtain mesenchymal marker expression, and manifest a migratory phenotype [44,46,47]. The epithelial-to-mesenchymal transition is observed during embryogenesis (type I), whereas it also occurs during wound healing and tissue repair due to regeneration, fibrosis, and inflammation (type II). Type III represents tumorigenesis, which is associated with local cancer invasion and distant metastasis (it is an unregulated transition providing the infiltrative properties and metastatic potential of the tumor) [44,47]. The transition is induced by the repression of E-cadherin (a regulator of cell adhesion and polarity), which is forced by the activation of a group of transcription factors such as Snail1, Slug, ZEB1, ZEB2, E47, and E2-2 [46,48,49]. The activation of these factors is due to the activation of the following pathways: transforming growth factor, tyrosine kinase receptors, and/or Wnt [46,48,49]. Interestingly, the epithelial-to-mesenchymal transition can lead to the dissemination of cancer cells into the blood vessels, where they are identified as circulating tumor cells in many women with malignancies. The presence of these cells is related to an elevated risk of tumor recurrence and distant metastases [44,50]. Cuevas et al. conducted a fundamental study that supported the conversion theory [51]. The authors observed the significance of the tumor suppressor Fbxw7 in the development of endometrial cancer using a genetic model system. They observed that the inactivation of Fbxw7 and PTEN led to endometrioid intraepithelial neoplasia (EIN) and well-differentiated endometrioid adenocarcinomas.

Subsequently, all of the patients with endometrioid adenocarcinoma developed ECS. The Fbxw7 mutations synergize with the deactivating mutations of PTEN and TP53, thus activating the epithelial-to-mesenchymal transition in well-differentiated endometrioid adenocarcinoma. This study showed that Fbxw7 loss is a key driver of the epithelial–mesenchymal transition and strongly supported the epithelial origin of ECS [51]. Osakabe et al. conducted an immunohistochemical analysis of the epithelial-to-mesenchymal transition in ECS. The authors investigated the expression of E-cadherin and EMT-related proteins (SNAI2, ZEB1, and TWIST1) through immunohistochemistry in the carcinomatous, transitional, and sarcomatous regions of ECS. The sarcomatous components had higher expression scores for ZEB1 and SNAI2 than those in the epithelial regions. The authors concluded that their results demonstrate that the EMT plays a significant role in the pathogenesis of ECS [52].

However, it should be noted that some molecular studies support the collusion theory, as a small proportion of ECSs are actual collusion tumors since they are molecularly biclonal [53,54,55,56,57]. Collusion tumors should be accurately diagnosed, as in some instances, the prognosis may be more favorable compared to that for conversion ECSs in a similar stage [52]. Moreover, The Cancer Genome Atlas (TCGA) data supports the conversion and combination theories, as an analysis of the clonal architecture suggests that the two carcinosarcoma components essentially share critical driver mutations and suggest combination or conversion [58].

Nevertheless, the conversion theory is supported by genomic (mainly clonality analysis, tissue culture, and ultrastructural analysis), molecular, histopathological, immunohistochemical, and clinical evidence. The sections below will clearly describe this further evidence [47].

It should also be highlighted that ECS cells have phenotypic plasticity and the potential to undergo both epithelial-to-mesenchymal and mesenchymal-to-epithelial transitions and generate secondary epithelia. It is speculated that the epithelial-to-mesenchymal transition has a significant role in the progression to metastatic carcinomas and further implicates the mesenchymal-to-epithelial transition during the subsequent colonization process. Therefore, the mesenchymal-to-epithelial transition facilitates the development of secondary tumor formation as disseminated cells have to shed their mesenchymal phenotype [7,59,60]. The conversions between cell lines prove the aggressive clinical features of ECS [7].

Grossly, a typical examination of ECSs reveals bulky, necrotic, and hemorrhagic sessile or polypoid masses, which often fill the entire endometrial cavity and prolapse through the cervix. These tumors are generally larger than endometrial adenocarcinomas due to their increased cellularity and sarcomatous differentiation. Deep myometrial invasion, cervical involvement, and extension beyond the uterus are often observed [36,61,62]. A tumor protruding through the cervix can mimic a cervical malignancy. The consistency of the tumor is usually soft, although it can be hard if the tumor consists of a significant proportion of cartilage or bone tissue [2,61,62,63]. Rarely, they can arise in an existing uterine polyp [61,62]. Occasionally, this tumor may fill the entire pelvic cavity, in which the uterus cannot be identified. The size of the tumor varies from 2 cm to 20 cm. The cut surface of the tumor reveals soft and moist tissue with significant areas of necrosis, hemorrhage, and cystic degeneration [23,36]. The gross characteristics of ECS are shown in Figure 1.

ECSs are biphasic tumors consisting of high-grade malignant epithelial and mesenchymal components with sharp demarcation. However, in some cases, this demarcation can be blurred. Additionally, one of the elements can represent a small proportion of the tumor, and extensive sampling is required to avoid classifying tumors either as pure uterine sarcoma or carcinoma. Rarely, the sarcomatous and epithelial components are low-grade tumors [64]. The epithelial component is usually the dominant part and most commonly consists of grade 3 endometrioid or serous carcinoma. The latter represents the most frequently observed epithelial component among patients with ECS [2,62,65].

Nevertheless, one study reported a high incidence (90%) of endometrioid components when the tumor represented a polypoid mass. The authors also observed a higher percentage (53%) of non-endometrioid histology in this series. The authors also speculated that women with polypoid ECS (showing a lesser extent of myometrial and lymphovascular space invasion) could have increased an overall survival compared to that in those with nonpolypoid tumors [66]. The epithelial part can also be transparent cell, clear cell (Figure 2D), undifferentiated, dedifferentiated (Figure 2A), and, rarely, squamous (Figure 2C) or mucinous adenocarcinoma. In the majority of cases, the epithelial part of the tumor consists of one epithelial type (72%); however, in 28% of cases, an admixture of two or three components can be observed [2,62]. Occasionally, neuroectodermal, neuroendocrine, melanocytic, and immature teratoid-like differentiation can be found in the tumor [61,62,67,68,69,70]. ECS with a mesonephric-like carcinomatous component has recently been described [71]. Endometrial hyperplasia or endometrial intraepithelial carcinoma (EIN) can be identified at the endometrium adjacent to the malignancy [61]. ECSs with different endometrial components and differentiation are shown in Figure 2.

The sarcomatous component may be either homologous (endometrial stromal sarcoma, fibrosarcoma, leiomyosarcoma) or heterologous (rhabdomyosarcoma, chondrosarcoma, osteosarcoma, liposarcoma) regarding whether the sarcomatous components resemble the uterine tissue or not (Figure 3). The most commonly observed homologous sarcomatous elements are high-grade stromal sarcoma (Figure 2B) and undifferentiated sarcoma, whereas the most frequent heterologous mesenchymal components are rhabdomyosarcoma and chondroid differentiation (Figure 2D). The incidence of heterologous differentiation varies between 25% and 50% [2,72]. However, one of the most prominent studies found approximately 41% heterologous sarcomatous differentiation among 906 women with ECS [73]. Similarly to the epithelial components, the tumor consists of one sarcomatous component in 67% of cases and two or more in 33% of women [2,73]. The percentages and types of carcinomatous and sarcomatous components should be part of pathology reports, as serous/clear cell histology and heterologous sarcomatous components are considered adverse prognostic factors in early-stage ECS [62]. Deep myometrial invasion (into the outer half of the myometrium) and lymphovascular space invasion are found in 32–45% and 38–64% of cases, respectively [4,36,46,74,75]. Regarding the metastatic pattern of ECS, the majority of metastases contain the epithelial component (90%), and lymphatic and peritoneal dissemination routes are most commonly observed [4,64,66,76]. The sarcoma component is related to the locoregional spread of the tumor (into the cervix, vagina, or fallopian tubes). Only the sarcoma components may occasionally be observed in distal metastases [77].

Immunohistochemistry is of limited value to diagnosis, although it may be helpful for identifying components that cannot be identified well in hematoxylin–eosin sections [61,62]. The conversion theory is supported by the concordance of the p53 staining between both (epithelial and mesenchymal) components of ECS. The expression of the p53 protein is either positive or negative in both components [61,78]. As ECSs are generally high-grade tumors, they express low levels of hormone receptors (hormone receptors in almost 25% of ECS cases). Estrogen is described in 20–30% of ECS cases, while the α isoform is more frequent within the carcinomatous component, and the β isoform is more often detected in the sarcomatous. One study found infrequent expression of estrogen receptor-a (8%) and -b (32%) and progesterone receptor-A (0%) and -B (23%) among 40 cases of ECS [79].

A differential diagnosis includes the following malignancies: low-grade adenosarcoma, endometrioid adenocarcinoma with spindle cell elements, dedifferentiated endometrial carcinoma, and endometrial adenocarcinoma, which contains benign heterologous elements. It is imperative to mention that ECS should be permanently excluded in cases of primary pure undifferentiated sarcoma or pleomorphic rhabdomyosarcoma. For such a diagnosis, extensive specimen sampling is recommended [61,62]. Differential diagnosis between ECS and carcinosarcoma of the uterine cervix is also recommended.

To date, there are no case studies or any data that have reported synchronous endometrial and ovarian carcinosarcoma.

## 5. The Molecular Landscape

Currently, The Cancer Genome Atlas (TCGA) has classified four novel molecular endometrial cancer subgroups: (I) POLE/ultramutated (POLE-mutated); (II) microsatellite-instability/hypermutated (MSI-H)/mismatch-repair-deficient (dMMR); (III) copy number-high/TP53-abnormal(p53-abn); and (IV) copy-number-low/TP53-wild-type, which also refers to no specific molecular profile (NSMP) endometrial cancers [5,80,81,82]. However, the TGGA study only observed women with endometrioid and serous histotypes, and molecular classification regarding ECS is lacking [4,5,80]. A recent meta-analysis observed the genetic profile of ECS in 231 patients. The analysis revealed a high incidence of the TP53 subgroup (73.9%) among the patients with ECS, and the prevalence of the other subgroups were 5.3% for the POLE-ultramutated, 7.3% for the MSI-H subgroup, and 13.5% for NSMP [81]. Gotoh et al. reported a similar proportion of ECS among the TCGA subgroups [58]—the majority of studies reported a high prevalence of the TP53 mutation, from 62% to 91%. The TP53 mutation is the most commonly detected molecular alteration in ECS, while the POLE and MSI-H mutation subgroups are less common in ECSs than in endometrioid cancers [54,82,83]. However, it is imperative to mention that the percentage of TP53 mutations depends on the grading of the epithelial component of the tumor. For instance, one study which included a large number of ECSs with low-grade carcinoma cases showed a lower range of TP53 mutations (62%) [57], whereas another study that investigated a low number of ECSs with low-grade carcinoma cases reported a high rate of TP53 mutations (91%) [10]. These findings show carcinosarcomas’ heterogeneous nature, where the tumor’s mutational profiles are determined mainly by the carcinomatous component. The difference in molecular profile according to TGGA classification between endometrial endometrioid cancer, clear cell cancer, serous endometrial cancer, and ECS is shown in Table 1 [10,58,82,84,85,86,87,88,89,90,91,92,93,94,95]. On the contrary, some authors have reported that ECS is almost exclusively of the p53abn molecular subtype after the exclusion of mimics [96]. One study showed a high prevalence of the PTEN, KRAS, ARID1A, and PIK3CA mutations among women with ECS with endometrioid differentiation, whereas the TP53, PIK3CA, FBXW7, CHD4, and PPP2R1A mutations predominated among women with ECS with serous differentiation [97]. Another study reported a high prevalence of TP53 (86%), PIK3CA (34%), and FBXW7 (23%) among 168 patients with ECS. Interestingly, HER2/neu (ERBB2) was encountered in 9% of tumors [98]. McConechy et al. also observed heterogeneous molecular features in ECS. They divided it into two subgroups according to the mutation profile—endometrioid-carcinoma-like (frequent PTEN, ARID1A, PIK3R1, and POLE mutations) and serous-carcinoma-like (TP53, PPP2R1A, EP300, and FBXW7 mutations) [9]. However, serous-like mutations (TP53 (60–91%), FBXW (10–44%), PPP2R1A (15–30%), and HER2 (9–18%)) are more frequently observed compared to endometrioid like mutations (ARID1A (10–25%), KRAS (10–15%), PTEN (10–50%), and PIK3CA (20–40%)) [1,5,7,9,82,98,99,100]. In particular, TP53 mutations and FBXW7 are the genes that are more frequently mutated in ECS than in endometrial cancer [51]. The mutation rates for the aforementioned gene mutations vary among the studies. The similarities in the mutation frequencies and profiles between serous endometrial cancer and ECS raise speculation that the dedifferentiation of serous endometrial cancer plays a role in the formation of ECS [10]. The transgenic expression of histones H2A and H2B in a serous endometrial cancer cell line induced and escalated the expression of the EMT, contributing to sarcomatous transformation [14,85]. One study reported prevalence of somatic BRCA1/2 mutations in 18% and 27%, respectively, of ECS cases [101]. The mutational profile of ECS compared to that of endometrioid cancer and other non-endometrioid subtypes is shown in Table 2 [10,58,82,84,85,86,87,88,89,90,91,92,93,94,95]. It should be emphasized that the revised FIGO classification of endometrial cancer based on molecular classification predicts the disease prognosis in ECS better than the previous version [102,103].

## 6. Clinical Manifestations

ECS presents similarly to other types of endometrial carcinomas. Therefore, distinguishing ECS from other uterine neoplasms based on clinical features is not possible. Patients usually present with pyometra and postmenopausal bleeding, which in 15% of cases is accompanied by pelvic pain. A protuberant tumor through the cervix is also observed among 15% of patients, and it is likely to contribute to pelvic pain. Irregular vaginal bleeding or bloody/watery discharge is often observed in young menstruating patients. Other non-specific symptoms include abdominal pain/swelling, gradual weakness, and symptoms related to an enlarged uterus/pelvic mass or metastatic disease. Urinary or gastrointestinal symptoms are often observed in the advanced stage of the disease [2,5,14,23].

## 7. Diagnosis

Approximately half of patients are diagnosed at an early stage of the disease, whereas the other half are diagnosed at an advanced stage [4,20]. A diagnosis is established through an endometrial biopsy or a biopsy of the protruding polypoid mass. It is imperative to mention that preoperative sampling techniques (aspiration biopsy, hysteroscopic biopsies, or dilatation and curettage) have a low diagnostic accuracy for ECS, irrespective of the sampling methods. Pham et al. retrospectively investigated the accuracy of the abovementioned techniques in 1273 patients diagnosed with ECS. The authors reported positive predictive values for hysteroscopic biopsy and dilatation/curettage of 73% and 78% and negative predictive values of 37% and 26%, respectively [104]. Additionally, in a small proportion of cases, an endometrial biopsy can only detect the carcinomatous or sarcomatous components of the tumor, and an accurate diagnosis is established in the final hysterectomy specimen [5,7]. Liquid biopsy is another innovative non-invasive technology which can be used as a diagnostic and prognostic test. It can detect circulating tumor DNA and circulating free DNA and provide important information on the genetic basis of cancers. Analyzing uterine lavage or any abnormal bleeding could help in detecting and studying proprotein convertases, which are associated with tumor development. Furthermore, proprotein convertases inhibitors could have a future role in ECS treatment [105].

Imaging modalities (abdominal/transvaginal ultrasound, magnetic resonance imaging (MRI), computed tomography, and positron emission tomography) are valuable tools for diagnosis and staging. One study reported that multiparametric MRI features may be a useful tool for distinguishing ECS from endometrioid adenocarcinoma [106]. Similar to endometrial cancer, ultrasound measurements of endometrial thickness and the Doppler parameters of the uterine arteries can also play a role in the prediction of ECS, as often the tumor fills the entire endometrial cavity [107]. Elevated perioperative serum levels of Ca-125 are associated with deep myometrial invasion and extrauterine diseases. Postoperative CA125 elevation is a negative prognostic factor for overall survival [108]. ECS is diagnosed at the advanced stages more frequently than other types of endometrial cancer. Approximately 60% of women initially diagnosed as in the early stage of the disease are upstaged after surgical evaluation. More than one-third of women (30–40%) present with a metastatic lymph node, and approximately 10% will have distant metastases, where the lung is frequently affected [1,5,82,100,104]. Therefore, the 5-year overall survival for women with locally advanced or metastatic disease is approximately 10–30% [15]. However, even women at the early stages of the disease experience recurrence in approximately 45% of cases, leading to a 5-year mortality of 50% [1,5,82,100,108].

## 8. Prognostic Factors

The prognostic factors for ECS can be divided into prognostic factors in the early and advanced stages, as some of them differ according to the stage of the disease. Negative prognostic factors particularly for the early stage of the disease include a tumor size > 5 cm; myometrial and cervical involvement; uterine rupture or perforation during hysterectomy; and inadequate staging. Other prognostic factors for the early stage of the disease are shown in Table 3 [1,3,4,5,41,47,61,62,63,76]. Unfavorable prognostic factors for the advanced stage of the disease are laparoscopic surgical interventions, suboptimal debulking, and discordance in HER2 status between primary and metastatic tumors. Regarding the molecular profile, TP53 mutations and a non-specific molecular profile of the tumor are associated with worse outcomes compare to those for POLE-mutated and MIS tumors [1,3,4,5,61,62].

However, a negative prognostic factor for both early and advanced disease is the presence of a heterologous sarcomatous component. Historically, heterologous sarcomatous components were believed to be related to a poor prognosis. Later, it was mentioned that an adverse prognostic factor is when the epithelial component is grade 3 endometrioid, serous, or clear cell, as this is associated with a high incidence of metastasis, lymphovascular space invasion, and deep myometrial invasion [3,61,62,76,79]. However, recent studies reported that the sarcomatous component is an independent risk factor for poor survival, mainly at an early stage of the disease [41,63,109]. Rosati et al. compared the overall survival between 60 women with ECS and homologous sarcomatous components and 35 women with heterologous sarcomatous components. All of the patients examined were in an early stage of the disease (I–II FIGO stages). In the multivariate analysis, the authors observed that only the heterologous components were significant negative prognostic factors associated with a decreased progression-free survival and overall survival [63]. A recent meta-analysis investigated the prognostic significance of heterologous components in carcinosarcomas of the gynecologic organs. A subgroup analysis showed that the heterologous components in ECS are associated with a worse overall survival, irrespective of the stage of the disease (the authors excluded multivariate analysis research, early-stage studies, ovarian carcinosarcoma studies, and studies with large patient samples) [72]. Another study reported that rhabdomyoblastic differentiation was an independent predictor of a worse 3-year overall survival [47]. Matsuo et al. reported that sarcomatous dominance (where the proportion of the sarcoma component exceeds more than 50% of the proportion of the carcinoma component), which is observed in 40% of ECS cases, is independently associated with decreased progression-free survival. This study’s multivariate analysis showed that homologous and heterologous sarcoma dominance patterns were independent prognostic factors for a decreased progression-free survival and site-specific survival compared to these values in homologous component dominance/non-dominance [41]. One study showed that lymphovascular space invasion with a sarcomatous component is independently associated with decreased progression-free survival and site-specific survival compared to these values in lymphovascular space invasion with the carcinomatous component [57]. LVSI with the sarcomatous component is often accompanied by sarcomatous dominance and heterologous sarcomatous components [57]. Smyth et al. retrospectively analyzed 77 patients with uterine carcinosarcoma and found that cervical stromal invasion was independently related to DFS and OS [110]. Similarly to endometrial cancer, the Eosinophil-to-Lymphocyte Ratio and the Eosinophil*Neutrophil-to-Lymphocyte ratio may also have prognostic significance in ECS patients. However, further prospective studies are needed to support this statement [111].

## 9. Treatment

It is imperative to mention that due to the rarity of the disease, all data regarding the treatment of ECS are mainly from retrospective studies and non-randomized data. Prospective studies are lacking, and the optimal treatment modalities are still debated. To date, no guidelines or standard consensus has been dedicated to ECS treatment. However, the standard of care for ECS aligns with that for other non-endometrioid high-grade tumors, as recommended by the ESGO/ESTRO/ESP, the National Comprehensive Cancer Network (NCCN), and the European Society of Molecular Oncology (ESMO) guidelines [8,75,81]. Generally, multimodal treatment is recommended, irrespective of the disease’s stage, due to the tumor’s aggressiveness [1,4,5].

### 9.1. Surgery

Surgical treatment in the early stage of the disease (stages I–II) includes a total hysterectomy with bilateral salpingo-oophorectomy, infracolic omentectomy, peritoneal biopsies, peritoneal cytology, and systematic pelvic/paraaortic lymph node dissection up to the level of the renal veins. Minimally invasive surgery is recommended in the early stages. However, intra-peritoneal tumor spillage, tumor rupture, and uterine morcellation should be avoided. Mini-laparotomy or the use of an endobag can be considered if vaginal extraction risks uterine rupture [8]. Metastasis beyond the uterus and pelvis (excluding lymph node metastases) is a contraindication for minimally invasive access [8]. A radical hysterectomy is only indicated for achieving free surgical margins, especially in cases of extracervical parametrial or vaginal involvement [1,4,5,8,81]. A radical hysterectomy in stage I–III endometrial cancer is not associated with improved progression-free survival and overall survival compared to those with a total hysterectomy [108]. Fertility-sparing surgery and ovarian preservation are contraindicated [8]. A hysterectomy with bilateral salpingo-oophorectomy and further surgical staging (irrespective of a woman’s age) is recommended in cases of endometrial polypectomy and ECS in the final pathology examination. The adnexa should be also removed in cases of hysterectomy without salpingo-oophorectomy for preoperative benign gynecological disease and ECS in the final pathology report [1,4,5,8].

Infracolic omentectomy is part of staging surgery regardless of the low incidence of microscopic metastasis in early-stage ECS—at approximately 6% (6 patients among 106) [112]. Positive peritoneal cytology is a negative prognostic factor, although it is not a cancer staging component [4]. Systematic lymph node dissection could be replaced by sentinel lymph node mapping using indocyanine green without compromising the oncologic outcomes [113]. The therapeutic effect of lymph node dissection is debatable [5]. In cases of unsuspected ECS in the final hysterectomy specimen and the absence of lymph node dissection, avoiding additional surgery with subsequent adjuvant chemoradiation is recommended [5]. In cases of hysterectomy without bilateral adnexectomy for benign uterus diseases and unsuspected ECS in the final hysterectomy specimen, additional surgery for the removal of both adnexa is recommended [1,4,8,81]. The surgical treatment for advanced disease includes open optimal debulking surgery with no macroscopic residual disease for patients with a good performance status. Only bulky retroperitoneal lymph nodes should be resected [1,4,5,8,81]. Suboptimal debulking is not associated with increased survival benefits compared to those under chemotherapy alone [8,75,81].

### 9.2. Chemotherapy

Chemotherapy plays a significant role in the management of patients with ECS. Neoadjuvant cisplatin-based chemotherapy followed by surgery can be initiated in rare cases with unresectable locally advanced disease and no evidence of disseminated disease [5,8,114]. A carboplatin/paclitaxel doublet regimen is considered the first-line treatment for early-stage or locally advanced-stage ECS. There is no standard of care in terms of the second-line options, although cisplatin/paclitaxel and ifosfamide/paclitaxel can be alternative regimes for women with hypersensitivity reactions to carboplatin [5,8,80,115]. However, data have shown that gemcitabine combination therapies are more effective than the carboplatin/paclitaxel regime, as this combination induced apoptosis in ECS-patient-derived organoids [116]. Chemotherapy alone is associated with an increased OS compared to that under observation, but avoiding radiotherapy could be related to an increased risk of local recurrence [117]. However, one study reported no impact on the overall survival after adjuvant chemotherapy, irrespective of the stage of the disease [118]. Another retrospective study reported an increased overall survival after adjuvant chemotherapy and radiation therapy in stage I of the disease compared to that under observation only. However, the authors reported no survival benefit after adjuvant treatment in cases without myometrial invasion [119]. An encouraging alternative for advanced ECS with peritoneal carcinomatosis could be the optimal cytoreduction surgery with hyperthermic intraoperative peritoneal chemotherapy (HIPEC) [120,121,122]. However, the benefit of oncological treatment with HIPEC cannot be estimated due to a small number of heterogeneous cases (mainly with a non-endometrioid histology) and the retrospective nature of the existing studies.

### 9.3. Radiotherapy

Neoadjuvant radiotherapy with or without chemotherapy followed by surgery may benefit cases with unresectable locally advanced disease and no distant metastases [123]. Most of the endometrial cancer treatment and consensuses guidelines do not recommend adjuvant radiotherapy alone [4,5,8,81]. It appears to decrease the risk of pelvic recurrence at the early stage of the disease, but there is a high risk of distant metastasis and no difference in the PFS and OS compared to those in an observation group [117]. Radiotherapy alone is not recommended as an adjuvant treatment for the advanced stage of the disease either, as the majority of women will probably experience recurrent disease [8,124]. Vaginal brachytherapy could be an alternative option to external beam pelvic irradiation [7]. One study reported no local and distal recurrence after adjuvant brachytherapy and chemotherapy in the early stages of ECS [125]. In retrospective data, Seagal et al. observed a decreased hazard of death after adjuvant multiagent chemotherapy and vaginal brachytherapy in women with stage I ECS [126]. These studies could envisage future recommendations of incorporating adjuvant vaginal vault brachytherapy into the stage I ECS treatment.

Furthermore, adjuvant brachytherapy could replace external beam radiotherapy in stage I of the disease. This scenario allows for additional external beam radiotherapy in patients with pelvic recurrence. Adjuvant radiotherapy could benefit metastatic lymph nodes, unknown nodal status, heterologous differentiation, and sarcomatous dominance [1,4,5].

### 9.4. Combined Treatment

According to the ESGO/ESTRO/ESP guidelines for the management of patients with endometrial carcinoma, patients with IA FIGO stage ECS and POLE mutations represent a low-risk group, and adjuvant combination or lone treatment can be avoided [8]. In all other stages of ECS, adjuvant therapy is mandatory [8,81]. Generally, most studies recommend combined treatment (radiotherapy and chemotherapy), delivered using either the sandwich or sequential method. A combination of external beam radiotherapy plus chemotherapy is associated with an increased OS compared with that under either radiotherapy or chemotherapy alone. Furthermore, regarding the different chemoradiation regimes, the “sandwich” method shows a longer OS compared to that under sequential chemotherapy–radiotherapy or radiotherapy–chemotherapy methods [1,5,7,127].

### 9.5. Recurrent Disease

Recurrences in ECS are observed in over half of patients despite multimodal treatment. Even the early stage of the disease is associated with a high rate of recurrence (>50% of women), of which approximately 80% are distant metastases. Tung et al. reported 84 (85.7%) women with disseminated recurrences and 14 (14.3%) patients with isolated recurrences among 98 patients with ECS recurrences [128]. However, local recurrences in the pelvis and the abdomen are more often the cause of death compared to distant metastases [1,2,4,73]. Nevertheless, disseminated recurrence is associated with a significantly worse survival-specific recurrence than that in oligometastatic disease [128]. The treatment for recurrence depends on the performance status of the patient, previous adjuvant treatment, and the type of recurrence (locoregional, distant, or oligometastatic). In cases of locoregional resectable recurrence (peritoneal or lymphatic relapse) or oligometastatic disease, surgery can be an option if a complete resection is performed. Additional external beam radiotherapy in patients with a locoregional relapse may be initiated only in patients who have undergone vaginal brachytherapy. Reirradiation of patients with recurrent disease is not recommended. Chemotherapy or immunotherapy is the treatment choice in cases of disseminated disease [5,128]. Recently, promising results have been shown in the detection of early recurrences in ECS patients with longitudinal circulating tumor DNA testing using tumor-informed assays [129].

### 9.6. Hormone Therapy

Hormone therapy in ECS is controversial, as a small proportion of these tumors have hormone receptors. However, hormone therapy may be used for disseminated disease and elderly patients with a poor performance status who cannot undergo chemotherapy [1,4,5]. Additionally, hormone therapy can be beneficial for the management of recurrent ECS [130,131]. A recent case study reported a durable partial response to pembrolizumab/lenvatinib and the aromatase inhibitor letrozole in a patient with recurrent ECS with estrogen receptor alpha (ESR1) gene amplification. As ESR1 gene amplification occurs in 7% of ECS cases, aromatase inhibitors may have a therapeutic effect in this cohort of patients [10,130]. Another case study reported no evidence of disease recurrence after 25 months of letrozole therapy for patients with recurrent hormone-positive ECS [131]. A phase 2 study (the PARAGON trial) showed that anastrozole had a clinical benefit (43%) for patients with estrogen receptor/progesterone receptor-positive metastatic ECS [132]. Interestingly, hormone therapy can be initiated in the neoadjuvant setting in ECS. Romano et al. reported a case of locally advanced hormone-positive ECS with lymph nodes and bone metastases. After ten months of letrozole treatment, the patients underwent a radical hysterectomy with bilateral adnexectomy. The hormone therapy was stopped 5 years after treatment, and the patient was free of the disease [133].

## 10. Novel Therapeutic Options

The efficacy of novel therapeutic regimes for ECS is understudied.

### 10.1. Immunotherapy

#### 10.1.1. Immune Checkpoint Inhibitors

Immunotherapy has become the standard of care after the failure of platinum-based chemotherapy in women with advanced or recurrent endometrial cancer. However, most trials have either investigated a small proportion of ECS patients or have not included ECS patients. Therefore, the efficacy of immunotherapy in ECS patients remains unclear.

Immunotherapy (particularly immune checkpoint inhibitors) is a type of cancer treatment that uses the body’s immune system to confront tumor cells. Immune checkpoint inhibitors have been used for the treatment of various malignancies—non-small-cell lung cancer, malignant melanoma, bladder and renal cancers, etc. [4]. Programmed death 1 (PD-1) is a T-cell inhibitory checkpoint molecule expressed in tumor-infiltrating T cells (Figure 4). PD-L1 is a checkpoint protein which is expressed by tumor cells. The binding of PD-L1 to PD-1 stops the T cells from destroying the tumor cells in the body. Inhibiting the union of PD-L1 and PD-1 using either anti-PD-L1 or anti-PD-1 immune checkpoint inhibitors reactivates the immune system and allows the T cells to attack the cancer cells [134,135,136]. No clinical studies or trials have investigated immune checkpoint inhibitors as first-line treatment agents in ECS. Immune checkpoint inhibitors in ECS patients are mainly indicated for treating metastatic or recurrent disease. Immune checkpoint inhibitors (pembrolizumab or dostarlimab) could be therapeutic in ECS, as PD-L1 and PD-1 expression is observed in 58% and 56% of ECS patients [137].

The GARNET trial reported that dostarlimab monotherapy demonstrates durable antitumor activity among patients with mismatch-repair-deficient and microsatellite-unstable–highly advanced or recurrent endometrial cancers [138]. The phase II KEYNOTE-158 study also reported the efficacy of second-line pembrolizumab in patients with noncolorectal high-microsatellite-instability/mismatch-repair-deficient cancer who experienced failure with prior therapy. Pembrolizumab demonstrates a clinical benefit with a high objective response rate of 30.8% and tolerability [139,140]. However, pembrolizumab as a monotherapy shows less effectiveness in patients with microsatellite-stable or pMMR disease [141]. The KEYNOTE-B21 phase III study observed the efficacy of adjuvant pembrolizumab (as a first-line treatment) or a placebo plus chemotherapy for newly diagnosed, high-risk endometrial cancer. The patients underwent surgery and had no evidence of disease postoperatively, with no prior radiotherapy or systemic therapy. This study showed that the combination of adjuvant pembrolizumab/chemotherapy increases the disease-free survival only in patients with mismatch-repair-deficient endometrial cancer tumors [142]. These studies showed that pembrolizumab shows less effectiveness as a nontherapeutic agent either in early or advanced high-risk pMMR endometrial cancer [138,139,140,141,142]. On the other hand, a randomized phase III trial showed that first-line chemotherapy plus pembrolizumab and pembrolizumab maintenance in patients with advanced or recurrent endometrial cancer are associated with a longer progression-free survival than that with chemotherapy alone regardless of MMR status or histologic findings [143].

Tumor-infiltrating lymphocytes (CD3, CD4, CD8, FOXP3, PD-1, and PD-L2) have increased expression in the sarcomatous component compared to that in the epithelial component. Therefore, immune checkpoint inhibitor therapy could benefit patients with a stronger sarcomatous component (sarcomatous dominance) [144].

#### 10.1.2. Immune Checkpoint Inhibitors with Multi-Targeted Tyrosine Kinase Inhibitors

Combining immune checkpoint inhibitors and multi-targeted tyrosine kinase inhibitors (imatinib, sunitinib, lenvatinib) may enhance the therapeutic efficacy of immune checkpoint inhibitors. Multi-targeted tyrosine kinase inhibitors inhibit tumor-associated macrophages, contributing to the EMT [1,4]. The efficacy of combining pembrolizumab and lenvatinib was reported by the KEYNOTE-146 study, which enrolled 108 women with advanced endometrial cancer regardless of their tumor microsatellite instability status. This study showed a promising antitumor response, although ECS was not part of the trial. However, as one-third of the study population had serous carcinoma, which shares similar molecular profiling to serous-like ECS, it could be envisaged that combination therapy could be a possible treatment regime for ECS [145,146]. The KEYNOTE-775 phase III study showed that the combination of second-line lenvatinib plus pembrolizumab (for patients with pMMR advanced endometrial cancer) is associated with increased overall survival, progression-free survival, and objective response rates versus those under chemotherapy [147]. Therefore, the European Medicines Agency (EMA) approved the combination of pembrolizumab and lenvatinib as a second-line treatment for patients with advanced or metastatic endometrial cancer, regardless of MMR status [5]. The role of immune checkpoint inhibitors as a first-line treatment regime is still debatable.

Only small retrospective and non-randomized studies or case series have reported the role of immune checkpoint inhibitors, either in combination with tyrosine kinase inhibitors or without, particularly for ECS. One retrospective study reported eight patients (all pMMR and 75% PD-L1-negative) with advanced ECS who received second-line pembrolizumab plus lenvatinib after cytoreductive surgery and chemotherapy with carboplatin plus paclitaxel. The authors observed an improved PFS and OS after implementing pembrolizumab plus lapatinib [148]. Another recent retrospective study reported five patients with ECS (one patient had FIGO stage II disease, and four had stage III disease) who were treated with surgery and chemotherapy. Lenvatinib plus pembrolizumab was used in all patients as a second-line regime. The PD-L1 status of the patients was not investigated. All patients were pMMR. The authors reported a 40% overall response rate, with a partial response in two women and stable disease in one. Two women experienced progressive disease. Interestingly, the two women with a partial response had TP53 and NOTCH3 mutations. The authors concluded that the responses to lenvatinib and pembrolizumab therapy differ among the molecular classification groups for patients with ECS [149]. For instance, patients with ECS and POLE mutations have a better prognosis than that for other molecular ECS groups, as POLE-mutated tumors are more immunogenic. Therefore, patients with pathogenic POLE mutations have better outcomes after treatment with immune checkpoint inhibitors [150]. Another clinicopathologic and molecular study examined 11 ECS POLE-mutated cases. Ten patients had no recurrences—only one patient (with subclonal POLE) presented with stage IV of the disease. This study demonstrated that POLE-mutated ECS has distinctive morphologic and immunohistochemical features compared to those of p53-abnormal ECS and could have notable prognostic differences [151].

Conversely, Hunt et al. observed no partial or complete responses among seven patients with advanced or recurrent ECS who underwent treatment with lenvatinib plus pembrolizumab [152]. Therefore, future larger and prospective studies are warranted to assess the efficacy of this regimen.

#### 10.1.3. PARP Inhibitors

As mentioned above, approximately one-fourth and one-third of ECS cases have somatic BRCA1 and BRCA2 mutations [90]. Therefore, ECS with somatic BRCA1/2 mutations may be sensitive to PARP inhibitors. However, only a few case reports have investigated the efficacy of PARP inhibitors in carcinosarcoma patients. Laurent et al. reported the clinical benefit with olaparib for a patient with FIGO stage IVB ovarian carcinosarcoma with a germline BRCA 1 mutation (inguinal lymph node involvement) [153]. Studies suggest that ovarian carcinosarcoma and endometrial ECS cell lines exhibiting homologous recombination deficiency and BRCA mutations are more sensitive to olaparib than homologous-recombination-proficient carcinosarcomas [154]. These studies showed that PARP inhibitors could be associated with favorable outcomes in selected patients. The phase II DOMEC trial investigated the efficacy and safety of combining immune checkpoint inhibitors (durvalumab) and PARP inhibitors in metastatic or recurrent endometrial cancer (14% of patients had carcinosarcoma histology). This study did not meet the predefined threshold of a 50% 6-month progression-free survival. The latter at 6 months was 34%. However, this study included a heterogeneous group of patients with relatively unfavorable characteristics.

Additionally, some patients benefit from a prolonged response. One particular patient with p53-abnormal ECS had a durable response for more than 17 months, although she previously experienced unfavorable disease control after primary surgery and chemotherapy. The combined treatment was well tolerated regarding adverse events [155].

#### 10.1.4. HER2-Targeting Agents

Approximately one-fourth of ECS cases involve human epidermal growth factor receptor 2 (HER2) gene overexpression. Rottmann et al. investigated the characteristics of HER2 expression/amplification in 80 women, of whom 65 and 15 had uterine and tubo-ovarian carcinosarcomas, respectively. Thirteen women (twelve uterine and one ovarian, 16%) were HER2-positive. The HER2 positivity rate, particularly for ECS, was 19%. Interestingly, all HER2-positive carcinosarcomas had a serous or a mixed carcinoma component. Notably, all ECS patients with an endometrioid, clear cell, undifferentiated, or neuroendocrine carcinoma subtype were HER2-negative [156]. Another recent study observed the molecular characteristics of ERBB2/HER2 gene amplification among patients with gynecologic malignancies. The authors reported a higher incidence of ERBB2/HER2 gene amplification in uterine serous carcinoma, clear cell carcinoma, and carcinosarcoma. The rate of ERBB2 amplification among ECS patients was 7.9% [157].

It should be stressed that the HER2 status can differ between the primary tumor and metastases. The primary HER2-positive tumor can be HER2-negative in the recurrent or distant metastatic tumor, although the reverse discordance (primary HER2-negative to relapse HER2-positive) has also been observed [152,154,158]. Discordance in the HER2 expression between primary and metastatic ECS is a poor prognostic factor [159].

To date, the NCCN guidelines have recommended HER2 immunohistochemistry testing for the eventual treatment for advanced-stage or recurrent ECS [74]. The ESGO/ESTRO/ESP and ESMO guidelines for the management of patients with endometrial carcinoma mention the application of paclitaxel and carboplatin with or without trastuzumab for HER2-positive serous endometrial cancer. This treatment strategy could apply to ECS, which shares similar molecular characteristics to those of serous endometrial cancer. A recent study investigated the efficacy of adding trastuzumab to carboplatin and paclitaxel in 280 patients with advanced HER2-positive serous endometrial cancer or ECS. The authors found a significantly increased median overall survival in group I (adding trastuzumab) compared to that in group II (chemotherapy only). Notably, the ECS patients experienced even longer survival benefits due to the addition of trastuzumab. However, the group I patients experienced more adverse side effects, of which left ventricular systolic dysfunction was the most severe [160].

The STATICE trial examined the efficacy of trastuzumab deruxtecan in patients with advanced or recurrent ECS expressing HER2 who were previously treated with chemotherapy. The authors reported that trastuzumab deruxtecan is effective in women with ECS regardless of HER2 status [160].

#### 10.1.5. Trophoblast Cell Surface Antigen 2 (TROP2)-Targeting Agents

TROP 2, tumor-associated calcium signal transducer 2 (Tacstd2), is mainly overexpressed in poorly differentiated endometrial adenocarcinoma and uterine serous carcinoma (65%). TROP-2 overexpression is associated with a worse prognosis and predicts disease recurrence [160,161,162]. Antibody–drug conjugates (ADCs) comprise a tumor-targeting monoclonal antibody conjugated to a cytotoxic payload. TROP2-targeting ADCs such as sacituzumab govitecan (humanized anti-Trop-2 antibody—active metabolite of irinotecan, a topoisomerase-I inhibitor) and hRS7 (humanized IgG1 monoclonal antibody) could represent novel therapeutic agents in ECS [162,163,164].

#### 10.1.6. RAF/MEK Inhibitors Combined with FAK Inhibitors

One case study series showed a good response to the combination of avutometinib (an RAF/MEK inhibitor) and defactinib (a focal adhesion kinase (FAK) inhibitor) among patients with ECS who harbored mutations in the RAS/MAPK pathway genes. The combination was associated with a longer survival compared to that under single-agent treatment [165].

#### 10.1.7. Epithelial–Mesenchymal Transition Target Therapies

By blocking the epithelial–mesenchymal transition, the epithelial component will remain dominant, and therefore, the standard therapies for endometrial carcinoma could be initiated. Additionally, negative prognostic factors such as sarcomatous dominance could be avoided [166].

## 11. Conclusions

ECS represents a rare biphasic epithelial carcinoma with the poorest prognosis among all histological types of endometrial cancer. ECS originates from the epithelial components of the tumor, which undergoes an epithelial-to-mesenchymal transition. Approximately half of patients are diagnosed at the advanced stage of the disease, whereas half of patients at the early stages experience recurrence. Despite multimodal treatment strategies, its prognosis remains dismal. As there are no new insights into the surgical treatment of ECS, new adjuvant target therapies are the latest promising agents. Adjuvant treatments based on the molecular classification of ECS could increase the DFS and OS. Genomic assessments of this neoplasm could allow physicians to avoid additional pretreatment in cases of FIGO stage I POLE-ultramutated ECS. Further prospective trials in light of ECS’s biology are urgently needed to increase these patients’ OS.

## Figures and Tables

**Figure 1 medicina-61-01156-f001:**
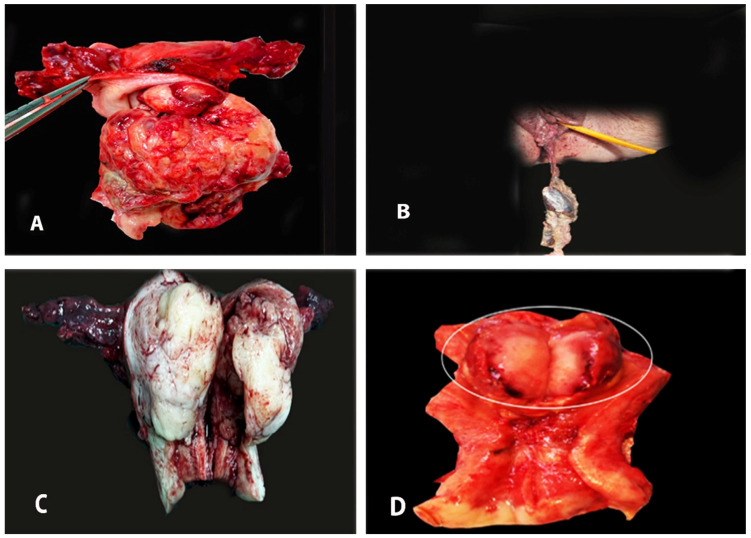
The macroscopic appearance of endometrial carcinosarcoma (the author’s own material). (**A**,**B**) ECS protruding through the cervix. (**C**,**D**) Postoperative specimens of ECS on final histological examination.

**Figure 2 medicina-61-01156-f002:**
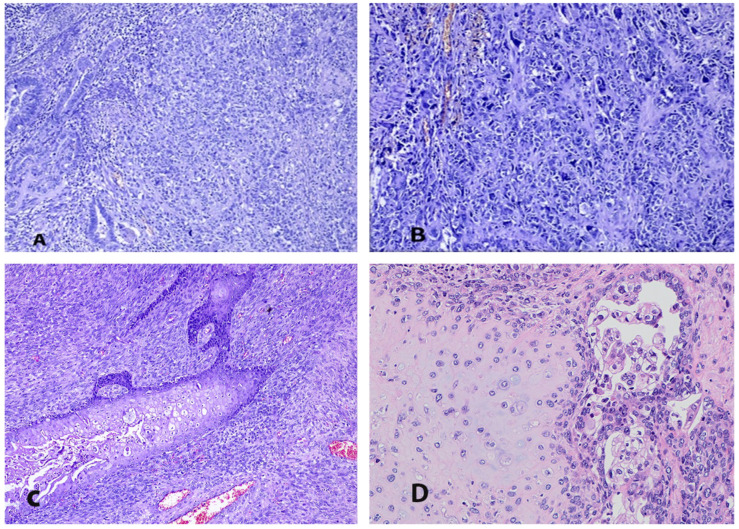
ECSs with different endometrial components and differentiation (the author’s own material). (**A**) Dedifferentiated epithelial malignant component and (**B**) high-grade stromal sarcoma in endometrial carcinosarcoma; (**C**) endometrioid carcinoma with extensive squamous differentiation and high-grade stromal sarcoma; (**D**) clear cell carcinoma with conventional chondrosarcoma.

**Figure 3 medicina-61-01156-f003:**
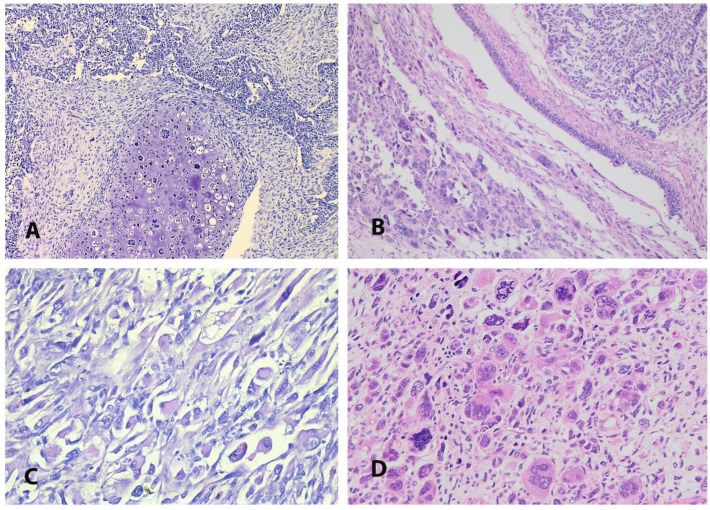
Heterologous sarcomatous differentiation in endometrial carcinosarcoma (the author’s own material). (**A**) High-grade endometrioid carcinoma with conventional chondrosarcoma; (**B**) high-grade endometrioid carcinoma and rhabdomyosarcoma with pleomorphic features; (**C**,**D**) high magnification of the rhabdomyosarcomatous component.

**Figure 4 medicina-61-01156-f004:**
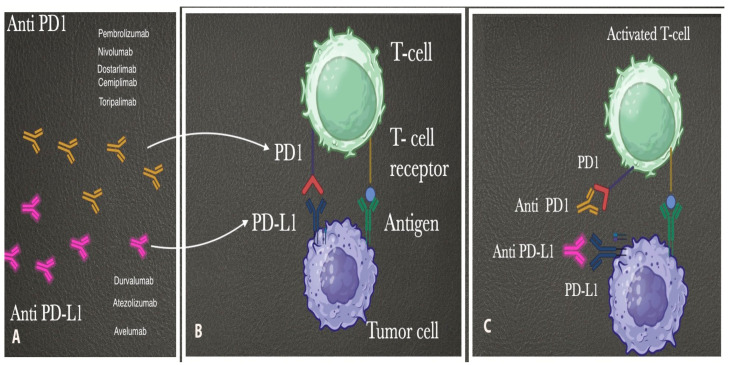
Immune checkpoint inhibitors (the author’s own material; the figure was obtained using BioRender.com). (**A**) Some anti-PD1 and anti-PD-L1 checkpoint inhibitors. (**B**) The T cell is inactivated. (**C**) Anti-PD1 and anti-PD-L1 checkpoint inhibitors block either PD1 or PD-L1. The T cell is activated and attacks the tumor cell.

**Table 1 medicina-61-01156-t001:** A comparison of the molecular profile of endometrial endometrioid, serous, and clear cell carcinoma and endometrial carcinosarcoma according to The Cancer Genome Atlas classification. MSI-H—microsatellite instability-hypermutated; NSMP—no specific molecular profile.

	Endometrial Endometrioid Cancer	Endometrial Carcinosarcoma	Serous Endometrial Carcinoma	Clear Cell Endometrial Carcinoma
**Molecular subtypes**				
*POLE-ultramutated*	7–9%	1.8–5.3%	2–7%	3.8–10.5%
*MSI-H*	15–28%	3.6–7.3%	2–11.8%	9.8–15.8%
*TP53 abnormal*	15–26%	73.9–91%	71–94%	42.5–57.9%
*NSMP*	39–55%	3.6–13.5%	0–5%	15.8–46%

**Table 2 medicina-61-01156-t002:** Mutational profiles of endometrial endometrioid cancer, endometrial carcinosarcoma, and clear cell and serous carcinoma.

Gene Mutation	Endometrial Endometrioid Cancer	Endometrial Carcinosarcoma	Serous Endometrial Carcinoma	Clear Cell Endometrial Carcinoma
p53-abn	7.5–21%	62–91%	71.1–88.4%	39.7–46%
ARID1A	32.5–54%	10–27%	6.7–9.3%	15.9–24%
PTEN	40–84%	18–41%	10–11%	7–21%
PPP2R1A	5–11%	10–28%	38–41%	15.9–36%
PIK3CA	10%	11–41%	22–35.6%	23.8–33%
FBXW7	17%	19–39%	24%	7.9–25%
SPOP	9.3%	3–18%	7–8%	14.3%
KRAS	24%	9–27%%	3%	7–14%
TAF1	17.1%	4–8%	4.7–13.5%	9.5%
PIK3R1	36–45%	11–23%	4.7–11%	15.9–18%
CTNNB1	34%	2–12%	1%	0%
ERBB2 (ampl.)	8%	9%	19%	6–11%
CHD4	9%	11–17%	18%	Not reported
CTCF	31%	7–17%	2%	Not reported

**Table 3 medicina-61-01156-t003:** The prognostic factors for ECS at the early stage of the disease. LVSI—lymphovascular space invasion; NSMP—non-specific molecular profile.

Prognostic Factors	Favorable	Unfavorable
**I.** **Early stage**		
**1.** **Tumor size**		
	<5 cm	>5 cm
**2.** **Surgical factors**		
	(−) Peritoneal cytology	(+) Peritoneal cytology
	Safe uterine removal	Uterine perforation or rupture
	Adequate staging (infracolic omentectomy, lymph node dissection, or sentinel node mapping)	Inadequate staging
	Ovariectomy	Ovarian preservation
	No fertility sparing	Fertility-sparing procedures
**3.** **Pathological factors**		
	(−) LVSI	(+) LVSI
	(−) LVSI within the sarcomatous component	(+) LVSI within the sarcomatous component
	Myometrial invasion < 50%	Myometrial invasion > 50%
	No invasion into the cervix	Cervical stromal involvement
	Endometrioid carcinoma component	Non-endometrioid carcinoma component (mainly serous-like)
	Homologous sarcomatous component	Heterologous sarcomatous component
	(−) Rhabdomyoblastic sarcomatous differentiation	(+) Rhabdomyoblastic sarcomatous differentiation
	(−) Sarcomatous dominance	(+) Sarcomatous dominance
	Low-grade carcinoma/low-grade sarcoma components	High-grade carcinoma/high-grade sarcoma components
**4.** **Molecular factors**		
	Pole mutation, MSI	TP53 mutation, NSMP
**5.** **Postoperative Ca-125 levels**		
	Normal	Elevated
**6.** **Adjuvant treatment**		
	Yes	No

## Data Availability

The authors declare that all related data can be made available to interested researchers by the corresponding author via email.

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
