# Peer review of "Carcinosarcoma of the Endometrium—Pathology, Molecular Landscape and Novel Therapeutic Approaches"

_medicina, 2025, doi:10.3390/medicina61071156_

Round 1

Reviewer 1 Report

Comments and Suggestions for Authors 1. [I. Introduction] The subsection is brief, but that's okay for this rare form of cancer. 2. [II. Epidemiology] Ok. 3. [III. Risk factors] This section contains too much unnecessary information. It could be shortened to one paragraph by combining the subsections. 4. [IV. Pathogenesis and epithelial to mesenchymal transition ] and [V. Pathology] sections could be merged. 5. [V. Pathology] Ok, but publisher's permissions are still required.  6. [VI. Molecular landscape] This section does not require separate Tables 1 and 2 and could be shortened. 7. [VII. Clinical manifestations] Illustrations required, if the authors have enough permissions. 8. [VIII. Diagnosis] Endometrial carcinosarcoma is a rare and aggressive form of cancer that affects the uterus. Traditional diagnostic methods include imaging, biopsy, and histopathological examination. However, analyzing menstrual fluid could provide a non-invasive diagnostic alternative or adjunct, especially for the endometrial stem cell subpopulations (CD90+, CD105+, CD73+, and CK7+)  [https://pubmed.ncbi.nlm.nih.gov/29397307/ ]. These approaches are in their early stages, but their non-invasive techniques could allow these studies to be referred to as liquid biopsies [https://pmc.ncbi.nlm.nih.gov/articles/PMC6896116/ ]. 9. [IX. Prognostic factors] The section's content is irrelevant because there are no logical connections with other subsections. The tables' content is unrelated to the main text. Essentially, the section is missing and needs to be written. 10. [X. Treatment] The presentation of thoughts and ideas is completely unacceptable. This section is a key part of the review as a whole and requires illustrations of appropriate quality and careful presentation. As it is, the therapeutic approaches are not discussed, so it is impossible to highlight promising areas and problematic issues. 11. [Figures]: - Figure 1 has no additional value, and should be erased. - Where is the missing "figure 3, second"? "Figure 3. Endometrial carcinosarcoma with myometrial invasion." - Figure 2 and 4 required note of the publisher permissions into the figure captions.

Reviewer 2 Report

Comments and Suggestions for Authors

Thank you for your great work. I have some concerns:

Risk factors of endometrial cancer should include diabetes, hypertension, early-aged menstrual… PMCID: PMC10025815

In diagnosis, the authors could mention the role of endometrial thickness and uterine artery Doppler parameters as soft markers for prediction of endometrial cancer in postmenopausal bleeding women. PMCID: PMC9483670

Please explain this point: “Fertility-sparing surgery and ovarian preservation are contraindicated [8].”

The authors could discuss the role of levonorgestrel intrauterine device in early-stage endometrial cancer among young women desiring the pregnancy.

The study could expand the discussion on Eosinophil-to-Lymphocytes Ratio (ELR) and Eosinophil*Neutrophil-to-Lymphocytes ratio (ENLR) in prognosing EC.  

The synchronous endometrial and ovarian carcinoma could be mentioned. 

Reviewer 3 Report

Comments and Suggestions for Authors

The title "Carcinosarcoma of the endometrium - pathology, molecular landscape and novel therapeutic approaches" is approprite for the article content. It summarizes to date knowledge on endometrial carcinosarcoma (ECS) and it underlines epithelial-to-mesenchymal transition in pathology, prognosis and treatment. The heterologous and non-endometrioid morphology of ECS components, TP53 mutation, lympho-vascular invasion, positive for cancer cells peritoneal cytology, suboptimal debulking and no adjuvant therapy worsen already dislmal prognosis. The article is interesting and it shows the need for advancements in ASC treatment.

However, below I listed some concerns:

1. "the worldwide annual incidence growth rate has increased by approximately 2 % " - to 2% or it grows 2% annually?

2. Figure 1 is rather unnecessary (too obvious).

3. Enlarge photograph in Figure 2B, please.

4. Figure 3 - merge the main caption "Figure 3" with A-D subcaption placed in various locations in the text.

5. The plain, white (not-blurred) separation between photographs in one figure seems to be more elegant, and Figs 2 and 3 could be similarly elegant like Fig. 4 further, Fig. 3 could be merged with Fig. 4 (it would contain eight photographs).

6. Tables 1 and 2 double simillar molecular change - "TP53 abnormal" and ""p53-abn" - in genes and protein expression, respectively.

7. "[56].dd LVSI with " - ?

8. "...with olaparib for stage IVB patients. (inguinal lymph node involvement) with germline BRCA1..." - is something missing here?

Comments on the Quality of English Language

1. "4.3 per 10 00000 for Afro-Americans vs. 1.7 per 1000000 for Caucasian women and 0.99 per 1000000 for women of other races [12]. In Europe, Boll et al. observed an increased growth rate of ECS from 5.1 to 6.9 per 1,000,00" - unify occurences to X per 100,000, please.

2. "(2-4% and undifferentiated (5%)" - close bracket, please => (2-4%)

3. "In 1906, Kheres proposed the term “ mixed mesodermal tumor” [23, 38]. In 1935, Mcforland observed 119 cases of ECS and tried to establish the gross and microscopic characteristics of the tumor. " - Mc Farland [38]

4. "Cuevas et al. conducted ..." - add [50] in the end of that sentence (not only further), please.

5. Delate unnecessary spaces in text, please.

6. "It should be emphasized that the revised FIGO classification of endometrial cancer based on molecular classification has better impact in predicting disease prognosis in ECS than the previous version" - correct "has better impact in predicting" (e.g. better predicts).

7. Table 3. "➢ 5 cm" correct to ">5 cm", please. Think of how to reduce empty spaces.

Round 2

Reviewer 1 Report

Comments and Suggestions for Authors

The authors have satisfactorily responded to most of my comments and made the necessary changes to the manuscript. However, another question occurred to me in reading the revised manuscript: Figures 1-3 are unpublished materials that were included in the review paper, but where are the patients' written informed consents?. Figure 4 is a diagram that should be referenced in the figure caption using the Biorender tool. The authors should clarify these points.

Author Response

Dear Reviewer,

Thank you for your insightful and constructive review of our paper dedicated on endometrial carcinosarcoma. We incorporated the recommended changes. All incorporated changes are highlighted by using the Track and Changes in Word.

Reviewer comments:

The authors have satisfactorily responded to most of my comments and made the necessary changes to the manuscript. However, another question occurred to me in reading the revised manuscript: Figures 1-3 are unpublished materials that were included in the review paper, but where are the patients' written informed consents?

Author’s Reply:

Thank you for the remark. We have mentioned at the end of the manuscript that informed consents was obtained from all the patients participating in the manuscript.

 Figure 4 is a diagram that should be referenced in the figure caption using the Biorender tool. The authors should clarify these points.

Author’s Reply: It is mentioned at the end of the manuscript the following sentence:

Figure 4 is assembled by BioRender.com (https://app.biorender.com/). We appreciate the use BioRender.com. Publication license number - VY28E17F76

We have even obtained a license number. This is the usual journal practice when you cite Biorender.

Additionally we incorporated the following sentence below the figure - Figure 4. Immune checkpoint inhibitors (Author’s own material. The figure was obtained by using BioRender.com).  

We are grateful for your valuable time and effort in reviewing our manuscript.

Based on your useful and scientific comments, we believe our manuscript has been improved to a higher level.

Reviewer 2 Report

Comments and Suggestions for Authors

Thank you for your responses. The paper is clear now. However, please check the reference. In this point:

Risk factors of endometrial cancer should include diabetes, hypertension, early-aged menstrual… PMCID: PMC10025815

Author’s Reply: We agree with the reviewer. Risk factors of endometrial cancer was briefly mentioned, as the main topic is endometrial carcinosarcoma.

The next text was inserted:

ECS share some similar risk factors with endometrial cancer – high body mass index, diabetes, hypertension, nulliparity, early-age menstruation, onset in post-menopausal age, and exogenous estrogen exposure [1, 4]. However, these risk factors are not specific, and their effects on ECS occurrence do not correlate significantly compared to Type I endometrial cancer [23].

The following reference was incorporated:

Nguyen, P. N., & Nguyen, V. T. (2022). Endometrial thickness and uterine artery Doppler parameters as soft markers for prediction of endometrial cancer in postmenopausal bleeding women: a cross-sectional study at tertiary referral hospitals from Vietnam. Obstetrics & gynecology science, 65(5), 430–440. https://doi.org/10.5468/ogs.22053

The above reference should be replaced by this reference:

Nguyen PN, Nguyen VT. Evaluating Clinical Features in Intracavitary Uterine Pathologies among Vietnamese Women Presenting with Peri-and Postmenopausal Bleeding: A Bicentric Observational Descriptive Analysis. J Midlife Health. 2022 Jul-Sep;13(3):225-232. doi: 10.4103/jmh.jmh_81_22. Epub 2023 Jan 14. PMID: 36950211; PMCID: PMC10025815.

Moreover, the list of reference with number [31] relating to the above paper was not inserted in the paragraph: “ECS share some similar risk factors with endometrial cancer – high body mass index, diabetes, hypertension, nulliparity, early-age menstruation, onset in post-menopausal age, and exogenous estrogen exposure [1, 4]. However, these risk factors are not specific, and their effects on ECS occurrence do not correlate significantly compared to Type I endometrial cancer [23].” Please check it.

Author Response

Dear Reviewer,

Thank you for your insightful and constructive review of our paper dedicated on endometrial carcinosarcoma. We incorporated the recommended changes. All incorporated

changes are highlighted by using the Track and Changes in Word.

Thank you for your responses. The paper is clear now. However, please check the reference. In this point:

Risk factors of endometrial cancer should include diabetes, hypertension, early-aged menstrual… PMCID: PMC10025815

Author’s Reply: We agree with the reviewer. Risk factors of endometrial cancer was briefly mentioned, as the main topic is endometrial carcinosarcoma.

The next text was inserted:

ECS share some similar risk factors with endometrial cancer – high body mass index, diabetes, hypertension, nulliparity, early-age menstruation, onset in post-menopausal age, and exogenous estrogen exposure [1, 4]. However, these risk factors are not specific, and their effects on ECS occurrence do not correlate significantly compared to Type I endometrial cancer [23].

The following reference was incorporated:

Nguyen, P. N., & Nguyen, V. T. (2022). Endometrial thickness and uterine artery Doppler parameters as soft markers for prediction of endometrial cancer in postmenopausal bleeding women: a cross-sectional study at tertiary referral hospitals from Vietnam. Obstetrics & gynecology science, 65(5), 430–440. https://doi.org/10.5468/ogs.22053

The above reference should be replaced by this reference:

Nguyen PN, Nguyen VT. Evaluating Clinical Features in Intracavitary Uterine Pathologies among Vietnamese Women Presenting with Peri-and Postmenopausal Bleeding: A Bicentric Observational Descriptive Analysis. J Midlife Health. 2022 Jul-Sep;13(3):225-232. doi: 10.4103/jmh.jmh_81_22. Epub 2023 Jan 14. PMID: 36950211; PMCID: PMC10025815.

Author’s Reply: It is done as recommended. The following reference was replaced by the one suggested.

Moreover, the list of reference with number [31] relating to the above paper was not inserted in the paragraph: “ECS share some similar risk factors with endometrial cancer – high body mass index, diabetes, hypertension, nulliparity, early-age menstruation, onset in post-menopausal age, and exogenous estrogen exposure [1, 4]. However, these risk factors are not specific, and their effects on ECS occurrence do not correlate significantly compared to Type I endometrial cancer [23].” Please check it.

 Author’s Reply: It was done as recommended. Sorry for the mistake. It was inserted. However, the mistake will be definitely seen during pdf approval if the article is accepted by the Editorial checking team of MDPI publishing house.

We are grateful for your valuable time and effort in reviewing our manuscript.

Based on your useful and scientific comments, we believe our manuscript has been improved to a higher level.